# Selective reinforcement of conflict processing in the Stroop task

**Arthur Prével** *, **Ruth M. Krebs, Nanne Kukkonen, Senne Braem**

Department of Experimental Psychology, Ghent University, Ghent, Belgium

* arthur.aeac@gmail.com

## Abstract

Motivation signals have been shown to influence the engagement of cognitive control processes. However, most studies focus on the invigorating effect of reward prospect, rather than the reinforcing effect of reward feedback. The present study aimed to test whether people strategically adapt conflict processing when confronted with condition-specific congruency-reward contingencies in a manual Stroop task. Results show that the size of the Stroop effect can be affected by selectively rewarding responses following incongruent versus congruent trials. However, our findings also suggest important boundary conditions. Our first two experiments only show a modulation of the Stroop effect in the first half of the experimental blocks, possibly due to our adaptive threshold procedure demotivating adaptive behavior over time. The third experiment showed an overall modulation of the Stroop effect, but did not find evidence for a similar modulation on test items, leaving open whether this effect generalizes to the congruency conditions, or is stimulus-specific. More generally, our results are consistent with computational models of cognitive control and support contemporary learning perspectives on cognitive control. The findings also offer new guidelines and directions for future investigations on the selective reinforcement of cognitive control processes.

## Introduction

To successfully navigate in a rich and changing environment, humans can rely on their ability to quickly respond in line with their goals in the face of conflicting events or changing task demands. This ability is often referred to as cognitive control [1–3]. In the lab, cognitive control is often studied with conflict tasks in which participants have to ignore task-irrelevant (dimensions of) stimuli and the associated responses. For example, in the classic color-naming Stroop task [4], participants are asked to name the ink color of color-words stimuli. Participants usually respond more slowly when the ink-color and word meaning are different (incongruent) than when they are similar (congruent), an effect known as the "congruency effect" caused by the habitual word-reading process.

In recent years, there has been an increasing interest in the interaction between cognitive control and motivation, and strong evidence has accumulated that cognitive control processes can be modulated by motivational signals [5, 6]. For example, in conflict resolution tasks,

**Data Availability Statement:** All data and codes files are available with description are available on OSF (https://doi.org/10.17605/OSF.IO/N7F45).

**Funding:** This study was supported by a starting grant of the European Research Council (ERC)

under the Horizon 2020 framework (grant No. 636116 awarded to RMK; and No. 852570 to SB).

**Competing interests:** The authors have declared that no competing interests exist.

several studies have shown that the presentation of reward cues or rewarded blocks (compared to no-reward cues or blocks) improves performance as reflected by faster and/or more accurate responses [7–10]. This effect of reward on cognitive control has also been reported in the context of other control functions, such as task-switching [11], response inhibition [12], and preparatory control processes [13], suggesting that people show increased cognitive control allocation to a task when reward incentives are presented. This beneficial effect of reward has also been integrated in recent theories on cognitive control. For example, assuming an inherent cost associated with the engagement in control processes, a recent model known as the Expected Value of Control (EVC) has proposed that the allocation and adjustment of cognitive control critically depends on a comparison between the reward prospect and the amount of (cognitive) effort needed to get that reward [14; see also 15]. Similarly, others have emphasized a learning perspective of cognitive control, according to which previous experiences learned through (reinforcement) learning processes guide the appropriate level of control that must be allocated to different contexts [16–18].

Understanding how reward can affect control representations and determine how cognitive control is allocated is relevant to predict how people will behave in different contexts or how we can intervene to improve control processes in specific disorders. However, most of the studies cited above focused on the anticipation of reward, using cues, stimulus features or block-wise manipulations that predict the future presentation of reward. These studies focused more on an invigorating/arousing effect of instructed reward prospect rather than on (reinforcement) learning components. For example, in a study by Krebs et al. [9], the prospect of reward for fast and accurate response was signaled to the participant by the ink-colors of the word stimuli, with certain items being associated with reward presentation and others not. Another common way to study the influence of reward on cognitive control is to present trial-by-trial pre-cues that signal reward for correct responses to the subsequent target [e.g., 10, 19]. Finally, other studies have manipulated reward prospect in cognitive control tasks in a block-wise manner [e.g., 20, 21]. In contrast, reinforcement learning studies traditionally focus on the differential effect of providing reward feedback to change specific dimensions of behaviors, e.g., response rate [22], response topography [23], response latency [24], or response variability [25].

The present study focused on the selective reinforcement of conflict processing, arguably one of the most extensively studied control functions, with important computational models developed to this end in the last 20 years [1, 2]. Extending the work of Krebs et al. [9], we tested whether it is possible to change conflict processing in participants by selectively rewarding responses following congruent versus incongruent trials in a Stroop task (for a similar reasoning in task switching, see [26]). To this end, we tested whether participants would show increased (decreased) conflict processing when reward is presented mostly following incongruent (congruent) items, as indicated by a smaller (larger) congruency effect.

This study also takes its inspiration from an extensive literature published in the associative learning domain that has demonstrated that action frequency (and the selection of a specific behavior more generally) increases when the action is consistently paired with a positive outcome, but decreases when the action-outcome contingency is degraded, i.e., when the outcome is presented in the absence of the action [27, 28]. Interestingly, previous studies in the cognitive control domain showed that reward effects on cognitive control depend on the instrumental contingency between the response and the expected reward [29–31]. The instrumental contingency between an action and an outcome (e.g., a reward) is commonly formalized by: $\Delta p = p(o|a) - p(o|\sim a)$, where $o$ represents the outcome presented and $a$ is the action performed [32–34]. Studies have extensively demonstrated that action frequency increases with $p(o|a) > 0$ and $\Delta p > 0$, and decreases with degraded or negative contingency, i.e., $p(o|\sim a) > 0$ and $\Delta p < 0$.

Here, we manipulated the probability of reward presentation separately on congruent and incongruent trials of a Stroop task. Namely, we considered $p(r|i)$ as the probability of reward when a participant performed a fast and correct response on an incongruent trial, and $p(r|c)$ as the probability of reward when a participant performed a fast and correct response on a congruent trial. We manipulated $\Delta p = p(r|i) - p(r|c)$ and tested whether conflict processing, measured by the size of the conflict effect, was modulated based on $\Delta p$. Specifically, we expected a smaller conflict effect for positive $\Delta p$ (i.e., reward is presented mostly following incongruent trials; named Incongruent Reinforced in our study) and larger conflict effect for a negative $\Delta p$ (i.e., reward is presented mostly following congruent trials; named Congruent Reinforced). The effect of instrumental contingency on conflict processing was tested in three experiments using a manual color Stroop task.

Experiment 1 tested the effect of instrumental contingency on conflict processing when reward is presented selectively on congruent and incongruent items, but keeping overall reward rate constant across conditions. Experiment 2 differed from Experiment 1 by the use of cues signaling the congruency identity of the next color-word stimulus, while Experiment 3 differed by the use of a fixed time-threshold for reward presentation. In all three experiments, participants were not instructed about the instrumental contingencies, as instructions are known to influence the effect of instrumental contingency on performance [35–37]. A preregistration for experiments 1 and 2 can be found on the Open Science Framework: https://osf.io/n7f45/?view_only=3d54d669405241cea21c4fbcc6409304. Experiment 3 was designed and conducted at a later stage to complement the results from experiments 1 and 2. Results found in Experiment 3 show that the size of the congruency effect can be modulated by selectively rewarding responses following congruent versus incongruent trials. However, our findings suggest in addition the existence of important boundary conditions.

## Experiment 1

### Participants

A total of 43 participants took part in the study in exchange for £7.50. Participants also received an additional bonus of on average £3.573 (range = 2.3–3.9, mean reward rate per trial in the test blocks = .213). One participant with below chance accuracy was excluded from the analysis and the data of 42 participants were included ($M_{age}$ = 26.595, SD = 5.315, Range = 18–35, 15 women, 27 men). They were recruited via Prolific and completed the experiment online. All participants were right-handed, with a normal color perception, normal or corrected-to-normal vision, English as first language, and no reported history of diagnosed mental disorders. The study was performed in accordance with the principles expressed in the declaration of Helsinki and with the protocol of the local ethics review board (Faculty of Psychology and Educational Sciences at Ghent University). Electronically signed informed consent was obtained from each participant before the experiment. They were informed of the testing time of approximately 60 minutes.

### Stimuli and procedure

All the material can be found on the Open Science Framework. The experiment was programmed and presented using JavaScript language and jsPsych (version 6.1.0) libraries. The experiment was run on desktop computers or laptops and required Chrome or Mozilla Firefox as browsers. Each trial of the color-naming Stroop task consisted of a word presented in the center of a black screen (BLUE, GREEN, RED, or YELLOW, letter font: Calibri; letter size: 60px; font-weight: bold). The ink-color of the word could be congruent (same) or incongruent (different) with regard to the word meaning (blue: RGB = [0, 0, 255], green: RGB = [0, 128, 0],

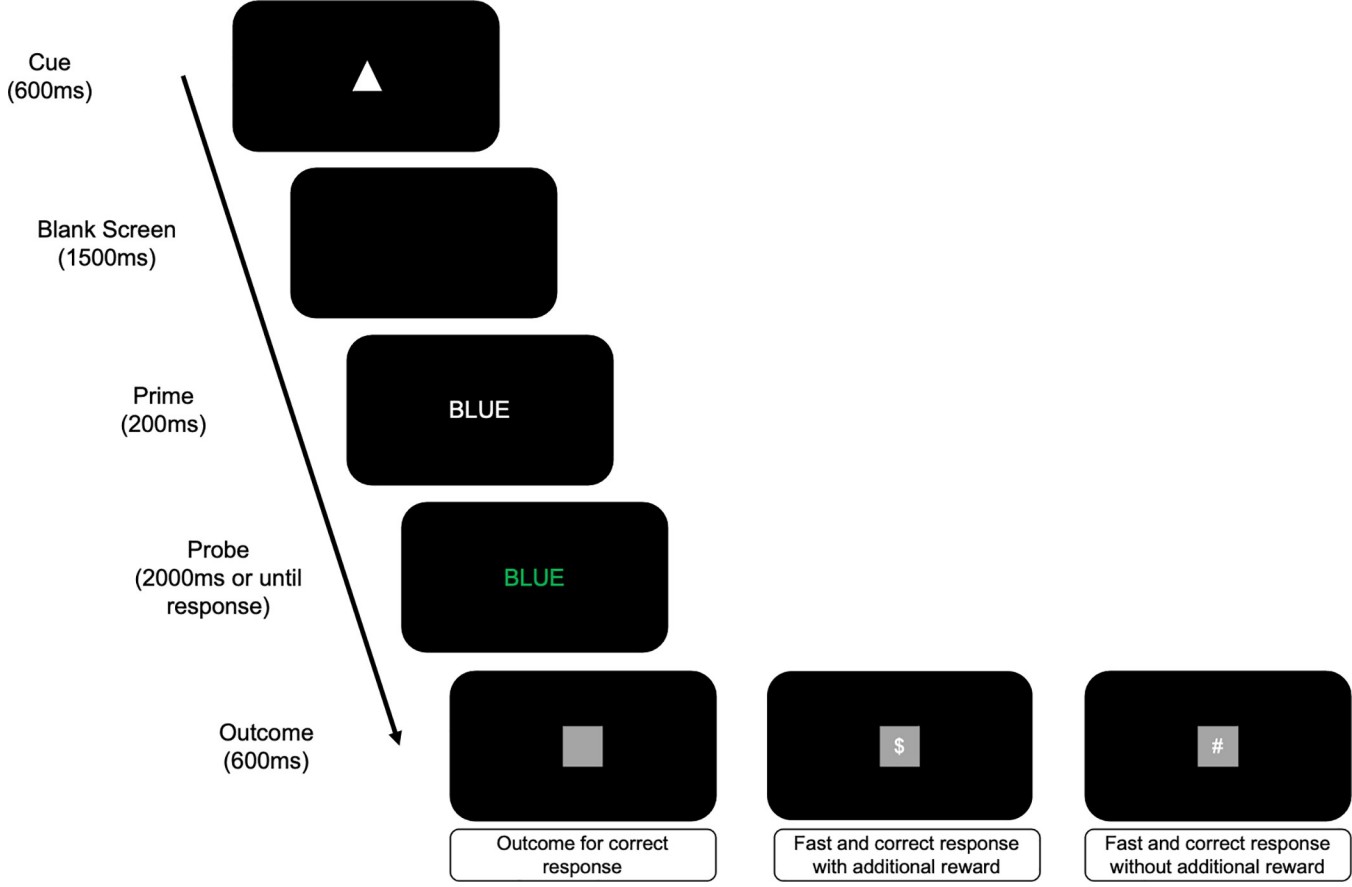

**Fig 1. Schematic illustration of a trial of the Stroop task.** A trial started with the presentation of a unique geometric shape for 600ms. This was followed by a blank interval for 1500ms, after which the word was presented in white for 200ms. Next, the word was colored and remained on screen for 2000ms or until the participant responded. Depending on the response performed by participants and the ongoing contingency, responses were followed either by a grey square, a grey square and the symbols "$" or "#", or a blank screen (i.e., no outcome) for 600ms. Trials were separated by a 600ms inter-trial interval.

red: RGB = [255, 0, 0], or yellow: RGB = [255, 255, 0]). On each trial, participants were asked to indicate the ink color of the word by pushing one of four keys and ignore the word meaning. Responses were collected via a keyboard using the keys "S", "D", "J", and "K". The response mapping of the ink-colors to the S, D, J, or K response keys was randomized across participants. We selected four incongruent trials (out of the twelve possible combinations) by combining the ink color assigned to the "S" key with the color word associated to "D" key, and vice versa, and the ink color assigned to the "J" key color word associated to the "K" key, and vice versa. Trial structure of the Stroop task is illustrated on Fig 1. A unique geometric shape (circle or triangle, randomized across participants; RGB = [255, 255, 255]) was presented for 600ms to signal a new trial to the participants. The same shape was used for the entire experiment. This manipulation differentiates Experiment 1 from Experiment 2, in which both shapes (circle and triangle) were used to signal the upcoming congruency identity (congruent versus incongruent) of the next color-word stimulus (see Experiment 2 for details). Shape presentation was followed by a blank interval for 1500ms, after which the word was presented in white (RGB = [255, 255, 255]; letter font: Calibri; letter size: 60px; font-weight: bold) for 200ms. This initial presentation of the (irrelevant) word stimulus was used to increase the impact of the irrelevant stimulus dimension, and hence the interference effect. After 200ms, the word was colored and remained on screen for 2000ms or until the participant responded. Depending on

the response performed by participants and the ongoing contingency, responses were followed by either a grey square, a grey square and the symbols "$" or "#", or a blank screen (i.e., no outcome) for 600ms (see below for details). Trials were separated by a 600ms inter-trial interval. The experiment consisted of 5 blocks: A baseline block without rewards; a second baseline block with contingent reward; and three experimental blocks in which the instrumental contingency between the responses performed and reward were manipulated (Congruent Reinforced, Equally Reinforced, or Incongruent Reinforced).

After reading the general study information and providing their informed consent, participants were instructed about the Stroop task and the color-response mapping (S1 Appendix). Participants then received four practice trials on congruent items, four practice trials on incongruent items, and eight mixed practice trials on both congruent and incongruent items. On these practice trials, a grey square was presented contingent to correct responses while incorrect responses or misses were followed by a blank screen. After the practice trials, participants performed a baseline block of 80 Stroop trials (10 presentations per item). Similar to the practice trials, correct responses were followed by a grey square and incorrect responses or omissions by a blank screen. The baseline block included breaks of ten seconds after 32 trials (however, participants were allowed to shorten the break and continue earlier by pressing "A" key). After these 80 trials, participants with accuracy below 75% were asked to perform the baseline block again. In total, five participants had to perform the baseline block twice.

Completion of the baseline block was followed by a second baseline block with contingent reward. This block consisted of 160 Stroop trials (20 presentations per item). Contingent to correct and fast responses, participants were presented with a grey square, overlaid with a "$" symbol, while correct but slow responses were followed by a grey square only. Prior to the block, participants were informed that each time the symbol "$" was presented, they earned £.02 as additional bonus money (S1 Appendix). Incorrect responses and misses were always followed by a blank screen. To determine whether a response might be considered as fast or slow, two response-time thresholds (one for congruent trials and one for incongruent trials) were calculated and updated for each new trial using a percentile schedule method [38]. The response-time threshold on a congruent (incongruent) trial consisted of the tenth fastest response from the last 20 correct responses performed by a participant on congruent (incongruent) trials. A new correct response had to be performed below that threshold to be considered as fast. This method allowed us to keep the reward rate constant across trial types (congruent and incongruent) and across participants, with a reward rate of around 50% due to the response-time threshold selected on each new trial. Importantly, while participants were informed about the opportunity of receiving reward during this block, the exact contingency between response and outcome (i.e., that only correct responses below a certain response-time threshold would be rewarded), was not described and hence uninstructed. Our aim was to measure the unique effect of instrumental contingency and reinforcement learning on conflict (learning) processing, and to prevent any (biasing) effect of instruction about instrumental contingency. The purpose of this block was to train participants on detecting the response-reward contingency.

The second baseline block with contingent reward was followed by three blocks of 160 Stroop trials each (20 presentations per item), with different response-reward contingencies for congruent and incongruent trials. Specifically, participants completed a Congruent Reinforced contingency block, with a probability $p(r|c) = 1$ to get a reward (r) for responding fast and correct to congruent items (c) and $p(r|i) = 0$ for responding fast and correct to incongruent items (i), an Equally Reinforced contingency block, with $p(r|c) = p(r|i) = .5$ and an Incongruent Reinforced contingency block, with $p(r|c) = 0$ and $p(r|i) = 1$. The global reward rate was constant across blocks but varied locally on congruent or incongruent items. Similar to the

second baseline block, reward feedback was signaled to the participants by a "$" symbol. Unrewarded correct and fast responses (e.g., fast responses to incongruent items performed during the congruent reinforced contingency block) were followed by a "#" symbol. Response-time-thresholds were calculated for each new trial following the same rule used in the second baseline block. The order of these three experimental blocks was counterbalanced across participants. Similar to the second baseline, participants were informed about the opportunity of receiving reward during these blocks, but they were not instructed about the specific response-reward contingencies on congruent and incongruent items. The experiment ended with the completion of the behavioral inhibition and activation scales (BIS / BAS; [39]). However, analyses of interindividual differences were considered exploratory, and did not reveal conclusive results in any of the three experiments. Data and overview of these correlations are made available via the Open Science Framework.

## Results and discussion

Analyses were performed using JASP version 0.14.1 [40]. Mean responses time (RT, in milliseconds) and mean error rates (in percentage) were calculated for each congruency condition and block type. Only correct responses slower than 150ms from color-word onset and within +/- 2.5 SDs were considered. We compared RTs and error rates between the three experimental blocks by means of a 3*2 repeated measures ANOVA, with Contingency (Congruent Reinforced, Equally Reinforced, Incongruent Reinforced) and Congruency (Congruent, Incongruent) as within-subject factors. Greenhouse-Geisser correction was applied where sphericity was violated. The threshold selected for significance was $p < .05$. The primary focus of our analysis concerned the presence of an interaction between Contingency and Congruency, as evidence of cognitive control adjustment to the contingency manipulated in the three experimental blocks. In a pre-registered exploratory analysis, we also evaluated how performance was influenced by time-on-task. Considering that the congruency-reward contingencies were not instructed, and these changed between blocks, it is possible that the detection of new contingencies took time. This could dilute an effect of contingency when averaging all trials across a block. For this analysis, each block was divided in two sub-blocks (1 and 2) and Time-on-task was considered as a within-subject factor. Thus, RTs and error rates were submitted to an additional 3*2*2 repeated measures ANOVA, with Contingency, Congruency, and Time-on-task (1, 2) as within-subject factors. Raw data, scripts, and processed data can be found on the Open Science Framework, for all the three experiments. Data from first and second baseline blocks, not presented here, are also available.

**Response time results.** Fig 2 (Left) shows the mean RTs of the three experimental blocks as a function of Contingency (Congruent Reinforced [CR], Equally Reinforced [ER], Incongruent Reinforced [IR]) and Congruency (Congruent, Incongruent). This analysis revealed a significant main effect of Congruency in the expected direction (F(1, 41) = 167.546, p < .001, $\eta_p^2 = .803$), a trend-level effect of Contingency (F(1.741, 71.374) = 2.970, p = .065, $\eta_p^2 = .068$) with shortest RTs in the congruent reinforced and longest RTs in the incongruent reinforced block. There was no interaction between Contingency and Congruency (F(1.879, 77.053) = 1.162, p = .316, $\eta_p^2 = .028$). The 3*2*2 repeated measures ANOVA revealed no interaction between Contingency and Time-on-task (F(1.756, 72.006) = .355, p = .675, $\eta_p^2 = .009$), and no interaction between Contingency, Congruency, and Time-on-task (F(1.976, 81.033) = 1.054, p = .353, $\eta_p^2 = .025$).

**Error rate results.** Fig 2 (Right) shows the mean error rates in the three experimental blocks as a function of Contingency and Congruency. The 3*2 analysis revealed a significant main effect of Congruency (F(1, 41) = 11.869, p < .001, $\eta_p^2 = .224$) with more errors for

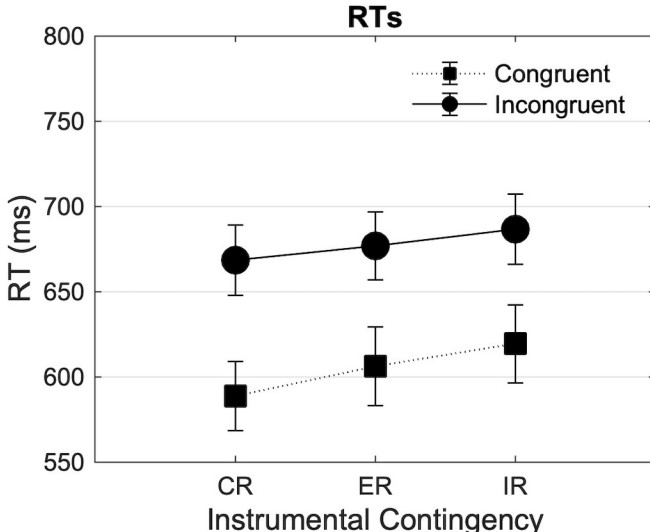
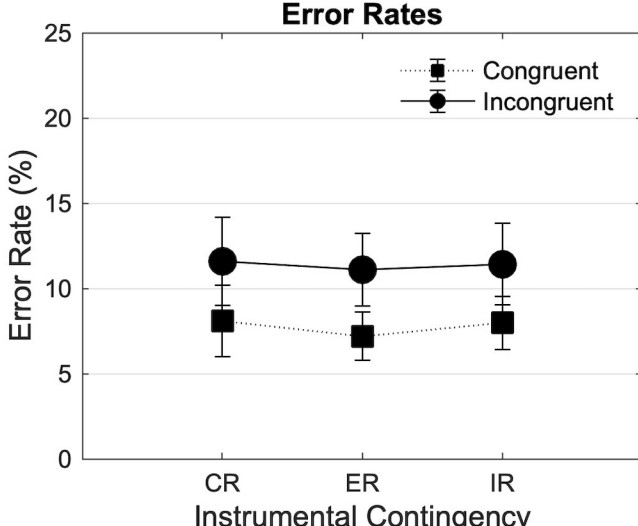

**Fig 2. Mean RTs (ms) and error rates (%) in Experiment 1 dependent on block and trial types.** Mean RTs (ms) and error rates found in experiment 1 are plotted as a function of Contingency (Congruent Reinforced [CR], Equally Reinforced [ER], Incongruent Reinforced [IR]), and Congruency (Congruent, Incongruent). Error bars represent SEM.

incongruent than congruent trials, but no main effect of Contingency ($F(1.609, 65.976) = .669$, $p = .485$, $\eta_p^2 = .016$), and no interaction between Contingency and Congruency ($F(1.990, 81.582) = .109$, $p = .896$, $\eta_p^2 = .003$). The 3*2*2 repeated measures ANOVA revealed no interaction between Contingency and Time-on-task ($F(1.575, 64.556) = 1.289$, $p = .277$, $\eta_p^2 = .030$), and no interaction between Contingency, Congruency, and Time-on-task ($F(1.807, 74.070) = .566$, $p = .553$, $\eta_p^2 = .014$).

**Interim discussion.** The RTs and error rates analyses revealed no significant interaction between Contingency and Congruency in the experimental blocks, nor a significant main effect of Contingency. The exploratory analysis on Time-on-task revealed no significant interaction between this factor and Contingency, or a three-way interaction with Contingency and Congruency. This means we found no support for our expected adjustments of congruency effects based on the differential congruency-reward contingencies. In preparing this study, we suspected that participants might be able to detect the instrumental contingencies but not to implement control due to the time constraints. The trend-level effect of contingency in the RT data with numerically faster responses in the negative compared to the incongruent reinforced contingency block could suggest that the contingency differences were picked up to some extent. Therefore, we conducted a second experiment which allowed participants to better prepare for the upcoming trial. Experiment 2 entailed the same manipulation as Experiment 1, but now with a cue presented at the beginning of each trial signalling the congruency condition of the upcoming stimulus (congruent or incongruent).

## Experiment 2

### Participants

A total of 43 participants took part in the study in exchange for £7.50. One participant with incomplete data was excluded and the data of 42 participants were included in the analysis ($M_{age} = 27.286$, SD = 5.558, Range = 18–35, 28 women, 14 men). Participants received an

additional bonus of on average £3.65 (range = 2.9–4, mean reward rate per trial in the test blocks = .219).

## Stimuli and procedure

The stimuli and procedure used in Experiment 2 were identical to Experiment 1, with the exception of cues presented at the beginning of each trial (see Fig 1) to signal the congruency identity of the upcoming color-word stimulus. Congruent (incongruent) word stimuli were always preceded by a square (triangle), with cue-congruency identity mapping randomized across participants. Participants were not informed about this cue-congruency identity mapping. Like Experiment 1, participants were not informed about the different response-reward contingencies manipulated on congruent versus incongruent trials in the experimental blocks.

## Results and discussion

The analysis was the same as the analysis performed in Experiment 1.

**Response time results.** Fig 3 (Left) shows the mean RTs in the three experimental blocks as a function of Contingency (Congruent Reinforced [CR], Equally Reinforced [ER], Incongruent Reinforced [IR]) and Congruency (Congruent, Incongruent). The 3*2 analysis revealed a significant main effect of Congruency in the expected direction (F(1, 41) = 165.086, p < .001, $\eta_p^2$ = .801), but no main effect of Contingency (F(1.924, 78.899) = 1.523, p = .225, $\eta_p^2$ = .036), nor an interaction between Contingency and Congruency (F(1.621, 66.443) = .693, p = .475, $\eta_p^2$ = .017). The 3*2*2 repeated measures ANOVA revealed no interaction between Contingency and Time-on-task (F(2, 82) = .248, p = .781, $\eta_p^2$ = .006), and no interaction between Contingency, Congruency, and Time-on-task (F(2, 82) = .999, p = .373, $\eta_p^2$ = .024).

**Error rate results.** Fig 3 (Right) shows the mean error rates in the three experimental blocks as a function of Contingency and Congruency. The 3*2 analysis revealed a significant main effect of Congruency in the expected direction (F(1, 41) = 17.349, p < .001, $\eta_p^2$ = .297), but no main effect of Contingency (F(1.829, 74.990) = 1.085, p = .338, $\eta_p^2$ = .026), and no

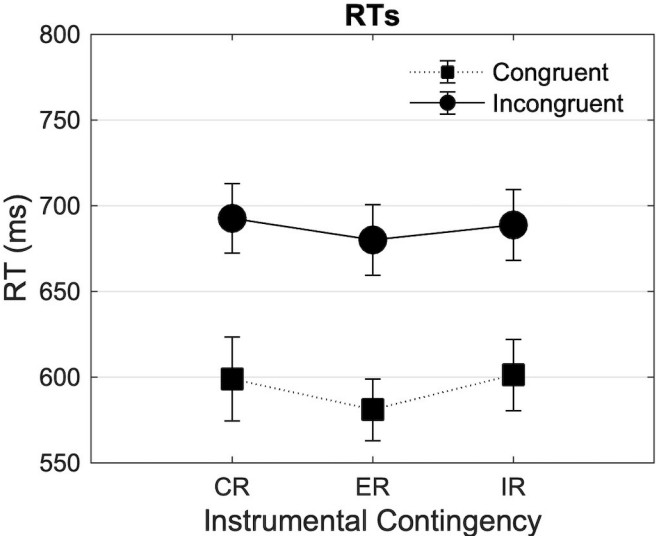
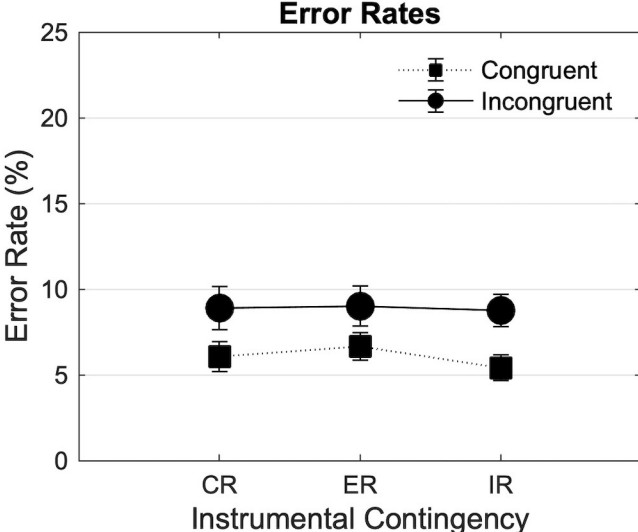

**Fig 3. Mean RTs (ms) and error rates (%) found in Experiment 2 across blocks and trial types.** Mean RTs (ms) and error rates found in experiment 2 are plotted as a function of Contingency (Congruent Reinforced [CR], Equally Reinforced [ER], Incongruent Reinforced [IR]), and Congruency (Congruent, Incongruent). Error bars represent SEM.

interaction between Contingency and Congruency (F(1.937, 79.412) = .782, p = .457, $\eta_p^2$ = .019). The 3*2*2 repeated measures ANOVA revealed no interaction between Contingency and Time-on-task (F(1.908, 78.244) = 2.716, p = .075, $\eta_p^2$ = .062), and no interaction between Contingency, Congruency, and Time-on-task (F(1.740, 71.336) = 1.046, p = .348, $\eta_p^2$ = .025).

**Interim discussion and across experiment analysis.** Experiment 2 was designed to test whether the presence of a cue signalling the identity of the next color-word stimulus would promote adjustments based on the different response-reward contingencies in the experimental blocks. We did not find any evidence for such differential adjustments and the trend-level effect of contingency on RTs found in Experiment 1 did not replicate. In addition, we did not find any evidence of an interaction between Contingency and Time-on-task, nor a three-way interaction with Congruency. Thus, while previous experiments demonstrated performance improvement triggered by rewards contingent on specific colors [e.g., 9], or by advance cues signalling the prospect of reward [e.g., 10], our findings from Experiments 1 and 2 do not provide evidence for such adjustments when selectively reinforcing conflict resolution (in form of condition-specific reward contingencies). In addition, the results highlight potential limits in the propositions made by recent models of cognitive control in which the engagement in control processes is regulated by a careful weighing of potential costs against gains [e.g., 14], or perspectives suggesting that instrumental learning should be seen as an important moderator of cognitive control processes [e.g., 16]. It is possible that reward can only have a global motivating effect on performance in cognitive control tasks, influenced by instrumental contingency [e.g., 29], or an effect that is specific to clearly identifiable features such as stimulus color [9], but that it is not possible to selectively reinforce conflict processing by congruency-reward contingencies. However, considering previous evidence of differential reinforcement in other control processes (task-switching; [26]) and the variety of behaviors and dimensions influenced by instrumental learning [41], the absence of performance adjustments based on congruency-reward contingencies could also be related to additional specific design features of the present experiment other than the absence of preparation suggested for Experiment 2. This would not be surprising considering that reinforcement learning and associative learning in general is known to occur in specific circumstances [see for example: 42, 43].

First, the instrumental reinforcement schedule used in the present experiments allowed to maintain globally a constant reward rate across blocks and participants of .50. However, given its adaptive nature (e.g., constantly adjusting the RT threshold based on the last 20 trials), participants might have felt unable to improve their performance. It is possible that participants adjusted control allocation and responses early in a block, until they detected and learned that their performance (in terms of RTs) did not have a global effect on reward rate. Using the rationale of the EVC model, the reinforcement schedule that we used might have implied that there was "no reason" for the participants to allocate more cognitive control, considering that the overall reward rate was kept at .50 independently of response speed. Second, this reward rate of .50 was perhaps too low to produce a significant change in performance across blocks. Third, considering the absence of instructions about the instrumental contingencies and the varying response-time threshold, it was potentially too difficult for participants to detect the exact congruency-reward contingency and therefore to adjust their response strategies. This is particularly true for the Equally Reinforced block during which learning the reward probability on congruent and incongruent trials probably interfered with learning what is the target response to perform to get additional reward.

To explore the possibility for an effect of the adaptive threshold in Experiment 1 and Experiment 2, we ran a post-hoc across-experiment analysis on the first half only, by means of a 2*2*2 repeated measures ANOVA, with Contingency (Congruent Reinforced, Incongruent

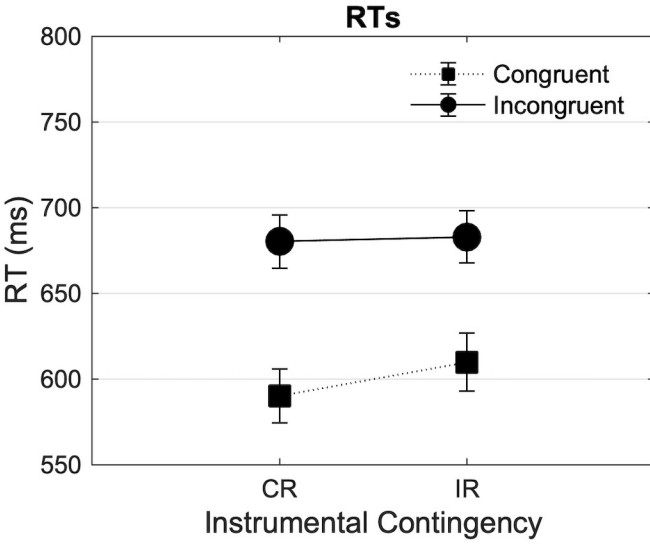
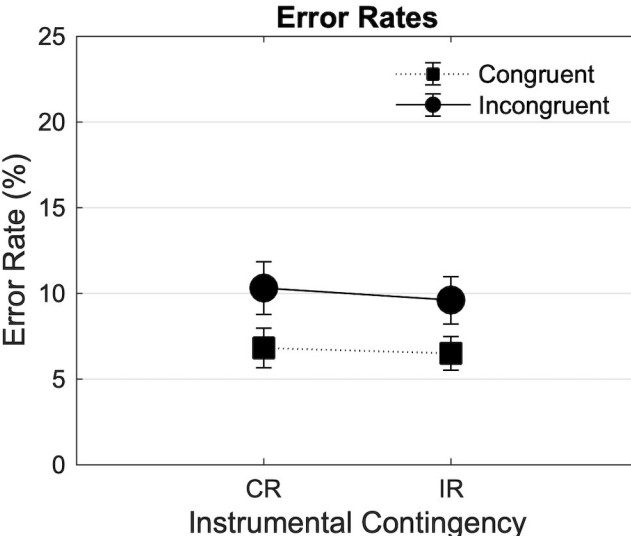

**Fig 4. Mean RTs (ms) and error rates (%) averaged across Experiments 1 and 2.** Mean RTs (ms) and error rates averaged across experiment are plotted as a function of Contingency (Congruent Reinforced [CR], Incongruent Reinforced [IR]) and Congruency (Congruent, Incongruent), and are calculated on the first sub-block. Error bars represent SEM.

Reinforced) and Congruency (Congruent, Incongruent) as within-subject factors, and Experiment (Exp 1, Exp 2) as between-subjects factor. We conducted this analysis considering only the first half of the Congruent Reinforced and the Incongruent Reinforced blocks, as we suspected a response adjustment in the expected direction early in a block, supposedly at a moment where participant did not yet detect that increased RTs had no global effect on reward. We focused on the Congruent Reinforced and Incongruent Reinforced blocks only as we expected the differences to be largest in those blocks. For completion, the same analysis was performed while adding Time-on-task as a within-subjects factor.

Fig 4 (Left) shows the mean RTs as a function of Contingency (Congruent Reinforced [CR], Incongruent Reinforced [IR]) and Congruency (Congruent, Incongruent), averaged across experiments (Exp 1 and Exp 2). The 2\*2\*2 analysis revealed a significant main effect of Congruency in the expected direction (F(1, 82) = 200.028, p < .001, $\eta_p^2$ = .709), and no main effect of Contingency (F(1, 82) = 1.790, p = .185, $\eta_p^2$ = .021). However, the analysis did reveal a significant Contingency by Congruency interaction (F(1, 82) = 4.490, p = .037, $\eta_p^2$ = .052), reflecting a larger difference between Congruent and Incongruent trials in the Congruent Reinforced compared to the Incongruent Reinforced block, and thus a larger congruency effect in the Congruent Reinforced block. In addition, the simple main effect of Contingency was marginally significant in Congruent trials (F = 3.840, p = .053), with longer RTs during Congruent trials in the Incongruent Reinforced block, but not in Incongruent trials (F = .092, p = .762). The analysis revealed no main effect of Experiment (F(1, 82) = .049, p = .826, $\eta_p^2$ < .001) and no three-way interaction between Contingency, Congruency, and Experiment (F(1, 82) = .157, p = .693, $\eta_p^2$ = .002), indicating that the above interaction pattern was similar across all two experiments. The analysis of Time-on-task in a 2\*2\*2\*2 repeated measures ANOVA revealed no interaction between Contingency and Time-on-task (F(1, 82) = .014, p = .907, $\eta_p^2$ < .001), a near-significant interaction between Contingency, Congruency, and Time-on-task (F(1, 82) = 3.516, p = .064, $\eta_p^2$ = .041), and no interaction between Contingency, Congruency, Time-on-task, and Experiment (F(1, 82) = .032, p = .859, $\eta_p^2$ < .001). This result is in line with our assumption that the effect of contingency on conflict processing was only effective in

the first half of the blocks due to the adaptive threshold used in Experiments 1 and 2. Finally, Fig 4 (Right) shows the mean error rates as a function of Contingency (Congruent Reinforced, Incongruent Reinforced) and Congruency (Congruent, Incongruent), averaged across experiments (Exp 1 and Exp 2). However, only the main effect of Congruency ($F(1, 82) = 18.726$, $p < .001$, $\eta_p^2 = .186$) proved to be significant in the expected direction (all other $p > .338$). The analysis of Time-on-task in a $2^*2^*2^*2$ repeated measures ANOVA revealed no interaction between Contingency and Time-on-task ($F(1, 82) = .387$, $p = .536$, $\eta_p^2 = .005$), no interaction between Contingency, Congruency, and Time-on-task ($F(1, 82) = 1.000$, $p = .320$, $\eta_p^2 = .012$), and no interaction between Contingency, Congruency, Time-on-task, and Experiment ($F(1, 82) = .019$, $p = .892$, $\eta_p^2 < .001$).

Thus, when only the first half of the Congruent Reinforced and the Incongruent Reinforced blocks is considered, our analysis suggests a response adjustment consistent with our expectations, i.e., a smaller congruency effect in the Incongruent Reinforced block in which reward is selectively presented on incongruent trials. This conclusion is supported in addition by the near-significant three-way interaction found when Time-on-task is added as a within-subject factor. Although speculative, we do believe this observation is in line with our assumption that the adaptive threshold (combined probably with a low reward rate) was responsible for the absence of a global adaptation of conflict processing in the expected direction. To address this issue, we conducted a third experiment in which the response-time threshold for congruent and incongruent trials was calculated based on performance in the baseline block and kept constant throughout the remaining blocks. This fixed response-time threshold is thought to help participants detect congruency-reward contingencies. At the same time, this approach likely leads to an increase in reward probability in each block, which might further motivate participants to optimize their performance. In a similar vein, we removed the equally reinforced block and only presented the congruent rewarded and incongruent rewarded congruency-reward contingency blocks–the most extreme conditions in terms of contingency learning.

Moreover, Experiment 3 featured additional small adjustments with respect to Experiments 1 and 2. First, participants were explicitly instructed that fast and correct response were required to receive additional reward. However, like in Experiments 1 and 2, no information was given about the differential contingencies in the congruent reinforced and incongruent reinforced blocks. Second, using causality ratings, we explicitly asked participants about their reward expectations following congruent and incongruent items separately, and per block. This procedure allowed us to test whether participants correctly detected these different contingencies and whether such awareness correlated with performance. Finally, Experiment 3 included four additional items (two more congruent and two more incongruent items) which were used as test items, often considered necessary to study adaptive changes in cognitive control [e.g., 44]. These test items were presented during the contingency blocks but with a similar reward rate across trial types, i.e., $p(r|c) = p(r|i) = .5$, and thus a different contingency than for items with contingency manipulations, now referred to as "learning items". The main purpose here was to test whether a performance adjustment on the learning items would generalize to test items, which can belong to the same, rewarded congruency condition, but have not been associated with differential congruency-reward contingencies themselves.

## Experiment 3

### Participants

A total of 47 participants took part in the study in exchange for £7.50. Five participants were excluded due to below chance accuracy or incomplete data, and the data of 42 participants

were included in the analysis ($M_{age}$ = 25.024, SD = 4.906, Range = 18–35, 29 women, 13 men). Participants received an additional bonus of on average £5.581 (range = 3.9–7.2, mean reward rate per trial in the test blocks = .354).

## Stimuli and procedure

The general procedures are identical with Experiments 1 and 2. Differences in design and stimuli are described in detail below. Responses were collected on six responses keys, using a keyboard with keys "S", "D", "F", "H", "J", and "K". In addition to the words "BLUE", "GREEN", "RED", and "YELLOW", presented in the experiments 1 and 2, the words "ORANGE" and "PURPLE" were also presented (letter font: Calibri; letter size: 60px; font-weight: bold), and ink-colors consisted in blue, green, red, or yellow used in previous experiments, as well as orange (RGB = [255, 165, 0]), and purple (RGB = [128, 0, 128]), for a total of six congruent items and six incongruent items. Response mapping of the ink-colors to the S, D, J, or K response key was randomized across participants. The trial structure was similar to Experiment 1 without pre-cues signaling the identity of the upcoming color-word stimulus. Thus, a unique geometric shape (circle or triangle, randomized across participants) was presented to signal a new trial to the participants, but not the upcoming congruency identity of the next color-word stimulus. Block structure was also similar except that Experiment 3 did not feature an Equally Reinforced contingency block.

After reading the instructions, participants received 12 practice trials on congruent items, 12 practice trials on incongruent items, and 12 practice trials on both congruent and incongruent items. After the practice trials, participants were tested in the baseline block on 84 trials of the Stroop task (7 presentations per items). Two participants had to perform the baseline twice because of poor level performance. Next, participants were tested on 156 trials of the second baseline with contingent reward (13 presentations per item). Contrary to experiments 1 and 2, participants were informed that a fast and correct response was required to receive additional reward. The two response-time thresholds, one for congruent trials and one for incongruent trials, were calculated based on performance in the baseline block, and consisted in the fifth fastest RT from the final 20 correct trials. These thresholds were kept constant throughout the remaining blocks. Finally, participants were asked two times during the second baseline block with reward (halfway the block and at the end) to rate to what extent a correct and fast responses in congruent and incongruent trials led to reward. The rating was indicated on a scale between -100 (a correct response during congruent/incongruent trials did not cause additional bonus money at all) to +100 (a correct response during congruent/incongruent trials absolutely caused additional bonus money). These ratings before the experimental blocks served to familiarize participants with the causality rating.

The rewarded baseline block was again followed by two experimental blocks of 240 Stroop trials each (including 20 presentations per item), one with a Congruent Reinforced and one with an Incongruent Reinforced contingency manipulation. In addition to the instrumental contingency manipulated on the original learning items (four congruent, four incongruent), four test items were included (two congruent, two incongruent) with similar probabilities within a block and across the two experimental blocks ($p(r|c) = p(r|i) = .5$). Similar to the rewarded baseline block, participants were asked to rate to what extent responding fast and correct during congruent or incongruent trials led to reward feedback two times per block (halfway the block and at the end). Participants were informed about the opportunity to obtain additional reward, but were not instructed about the differential congruency-reward contingencies, nor about the difference between learning and test items.

## Results and discussion

The procedure for data analysis was similar to Experiments 1 and 2 (see above), and was extended to the newly introduced test items. Additional analyses were performed on the causality ratings. Mean rating score was considered for each Contingency per Congruency conditions. Causality ratings were submitted to a 2*2 repeated measures ANOVA, with Contingency (Congruent Reinforced, Incongruent Reinforced) and Congruency (Congruent, Incongruent) as within-subject factors. We furthermore explored whether the size of the response adjustment across blocks (i.e., to what extent participants adjusted their performance between the two blocks) correlated with the causality ratings. As a measure of behavioral adjustment across blocks, we calculated for each participant the difference ($D_{CE}$) between the mean congruency effect on Congruent Reinforced contingency block ($CE_{CongR}$) minus the mean congruency effect on Incongruent Reinforced contingency block ($CE_{IncongR}$). As such, $D_{CE}$ reflects the extent of behavioral adaptation based on the different contingencies. In turn, $D_{CE}$ values were correlated with the causality ratings across participants. Causality ratings were also transformed to difference values ($D_{CR}$, $CR_{CongR}$ and $CR_{IncongR}$). $CR_{CongR}$ and $CR_{IncongR}$ were calculated for the Congruent Reinforced and Incongruent Reinforced block, respectively, by subtracting the mean CR for congruent trials from the mean CR for incongruent trials (i.e., reflecting the extent to which a participant expected incongruent stimuli to be followed more by reward than congruent stimuli). $D_{CR}$, on its turn, was the difference score between $CR_{CongR}$ and $CR_{IncongR}$. The relationship between (differences in) congruency effects and causality ratings was tested separately for learning and test items.

**Response time results.** Fig 5 (up Left, up Right) shows the mean RTs in the experimental blocks as a function of Contingency (Congruent Reinforced [CR], Incongruent Reinforced [IR]) and Congruency (Congruent, Incongruent), respectively for learning and test items. The 2*2 analysis on learning items revealed a significant main effect of Congruency in the expected direction (F(1, 41) = 207.157, p < .001, $\eta_p^2$ = .835), no main effect of Contingency (F(1, 41) = .594, p = .445, $\eta_p^2$ = .014), but a significant Contingency by Congruency interaction (F(1, 41) = 4.184, p = .047, $\eta_p^2$ = .093), reflecting a larger difference between congruent and incongruent trials in the Congruent Reinforced compared to the Incongruent Reinforced block, and thus a larger congruency effect in the Congruent Reinforced block. The simple main effect of Contingency in Congruent (F = 1.494, p = .229) and in Incongruent (F = .039, p = .844) trials was not significant. The 2*2*2 repeated measures ANOVA revealed no interaction between Contingency and Time-on-task (F(1, 41) = .108, p = .744, $\eta_p^2$ = .003), and no interaction between Contingency, Congruency, and Time-on-task (F(1, 41) = 1.328, p = .256, $\eta_p^2$ = .031). The 2*2 analysis on test items revealed a main effect of Congruency in the expected direction (F(1, 41) = 141.119, p < .001, $\eta_p^2$ = .775), but no main effect of Contingency (F(1, 41) = .259, p = .613, $\eta_p^2$ = .006), and no Contingency by Congruency interaction (F(1, 41) = .502, p = .483, $\eta_p^2$ = .012). The 2*2*2 repeated measures ANOVA revealed no interaction between Contingency and Time-on-task (F(1, 41) = 1.882, p = .178, $\eta_p^2$ = .044), and no interaction between Contingency, Congruency, and Time-on-task (F(1, 41) = .277, p = .601, $\eta_p^2$ = .007).

**Error rate results.** Fig 5 (middle Left, middle Right) shows the mean error rates in experimental blocks as a function of Contingency and Congruency, respectively for learning and test items. The 2*2 analysis conducted on learning items revealed a main effect of Congruency in the expected direction (F(1, 41) = 8.756, p = .005, $\eta_p^2$ = .176), no main effect of Contingency (F(1, 41) = .059, p = .809, $\eta_p^2$ = .001), and no interaction between Contingency and Congruency

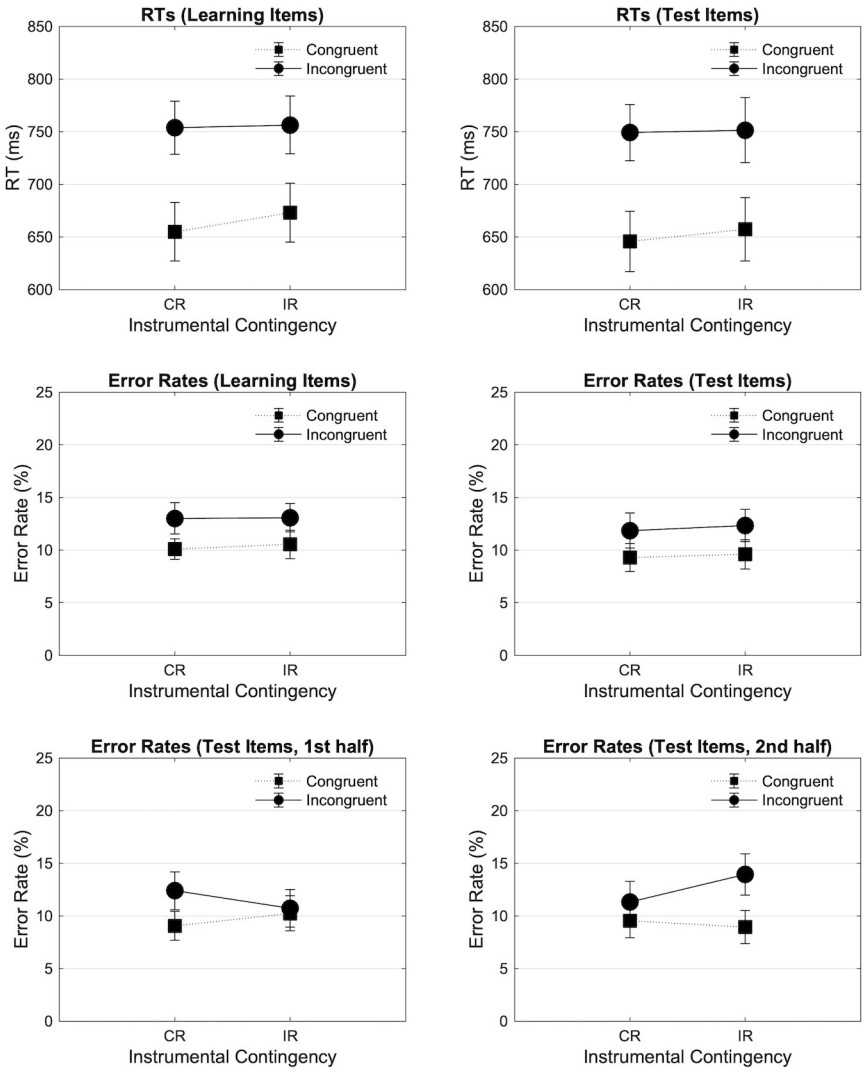

**Fig 5. Mean RTs (ms) and error rates (%) found in experiment 3 across blocks and trial types.** Mean RTs (ms) and error rates found in experiment 3 are plotted as a function of Contingency (Congruent Reinforced [CR], Incongruent Reinforced [IR]) and Congruency (Congruent, Incongruent), and across learning and test items. Error bars represent SEM.

$(F(1, 41) = .109$, p = .743, $\eta_p^2 = .003$). The $2^*2^*2$ repeated measures ANOVA revealed no interaction between Contingency and Time-on-task $(F(1, 41) = .124$, p = .726, $\eta_p^2 = .003$), and no interaction between Contingency, Congruency, and Time-on-task $(F(1, 41) = .983$, p = .327, $\eta_p^2 = .023$). Finally, the $2^*2$ analysis conducted on test items revealed a main effect of Congruency in the expected direction $(F(1, 41) = 5.651$, p = .022, $\eta_p^2 = .121$), but no main effect of Contingency $(F(1, 41) = .121$, p = .729, $\eta_p^2 = .003$), and no significant Contingency by Congruency interaction $(F(1, 41) = .025$, p = .875, $\eta_p^2 = .001$). The $2^*2^*2$ repeated measures ANOVA revealed no interaction between Contingency and Time-on-task $(F(1, 41) = .353$, p = .556, $\eta_p^2 = .009$), but a significant interaction between Contingency, Congruency, and Time-on-task $(F(1, 41) = 4.385$, p = .042, $\eta_p^2 = .097$), reflecting a numerically smaller congruency effect (on error rates) in the Incongruent Reinforced than in the Congruent Reinforced contingency block in

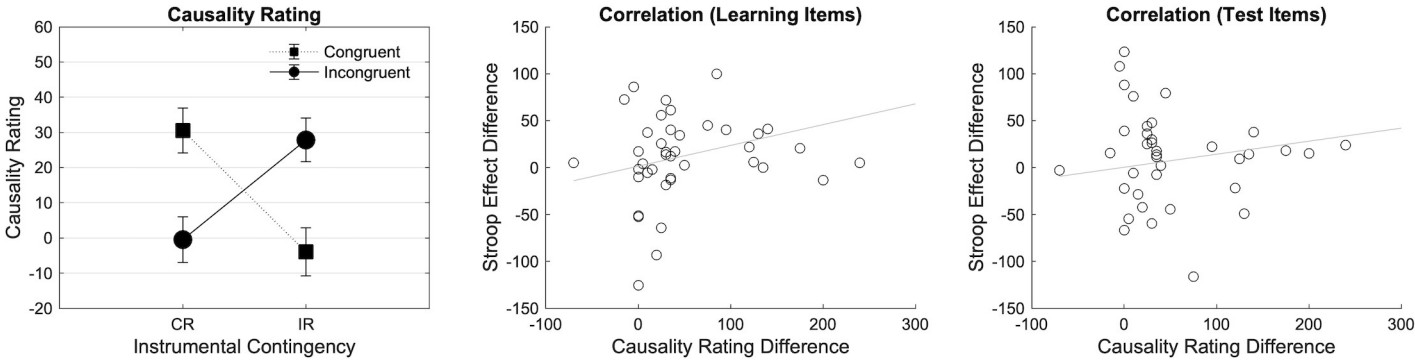

**Fig 6. Mean score for Causality Rating (CR) and Person's correlation on learning and test items in Experiment 3.** Mean score for causality rating (CR) found in experiment 3 is plotted as a function of Contingency (Congruent Reinforced [CR], Incongruent Reinforced [IR]) and Congruency (Congruent, Incongruent). Error bars represent SEM. Person's correlation between the difference score in causality ratings ($D_{CR}$) and the difference score in congruency effects ($D_{CE}$) on learning and test items is also plotted.

the first half (Fig 5 down Left), and inversely in the second half (Fig 5 down Right). However, none of the simple main effect of Contingency was significant (all p > .224).

**Causality rating and correlations.**   Fig 6 (Left) shows the mean score for causality rating as a function of Contingency (Congruent Reinforced [CR], Incongruent Reinforced [IR]) and Congruency (Congruent, Incongruent). The 2*2 analysis revealed no main effect of Congruency (F(1, 41) = .012, p = .912, $\eta_p^2$ = .000), no main effect of Contingency (F(1, 41) = .234, p = .631, $\eta_p^2$ = .006), but a significant interaction between Contingency and Congruency (F(1, 41) = 22.736, p < .001, $\eta_p^2$ = .357) with a significant simple main effect of Contingency in Congruent (F = 14.807, p < .001) and Incongruent (F = 9.415, p = .004) trials. In the Congruent Reinforced block participants perceived a stronger causality between congruent items and reward, while this was reversed in the Incongruent Reinforced block. In addition, Pearson's correlation showed a significant correlation between $D_{CR}$ (i.e., the difference score in causality ratings) and $D_{CE}$ (i.e., the difference score in congruency effects) on learning items (Fig 6 Middle; r = .388, p = .011), but not on test items (Fig 6 Right; r = .143, p = .366). This reflects that participants who adjusted response speed more as function of the different congruency-reward contingencies between the two blocks also perceived the congruency-reward contingencies in those two blocks more differently.

**Interim discussion.**   Experiment 3 was designed to test the effect of reinforcement learning on conflict processing by presenting reward selectively on congruent or incongruent items. This experiment differed from experiments 1 and 2 by the use of a fixed response-time threshold, and by removing the equally reinforced block. We found evidence of an adjustment of responses across blocks with a significant Contingency by Congruency interaction, and a smaller congruency effect in the Incongruent Reinforced compared to the Congruent Reinforced contingency block. This result is consistent with our expectation of enhanced (decreased) conflict processing when reward is selectively presented on incongruent (congruent) items, as indicated by smaller (larger) congruency effects. In addition, the analysis of the causality ratings (and the correlations with performance measures) showed that overall participants picked up the different congruency-reward contingencies, and that this explicit knowledge was related to performance adjustments in the different blocks. We found no evidence of such differential adjustment in accuracy. Moreover, we also did not find evidence for similar performance adjustments on the test items. We did observe a significant three-way interaction between Contingency, Congruency, and Time-on-task in the error rates, with a numerically

smaller congruency effect in the Incongruent Reinforced than in the Congruent Reinforced contingency block in the first sub-block, and this pattern reversed in the second sub-block.

## Discussion

Cognitive control refers to the ability of quickly responding to conflicting events or change in task demands [1]. It is well-known that motivation signals influence the engagement of cognitive control processes, as reflected by faster and/or more accurate responses [5, 6]. However, most of the previous studies focused on the invigorating effect of (instructed) reward prospect, rather than on (reinforcement) learning components. Extending our previous work on reward signals in conflict and task switching tasks [9, 26], the present study aimed to test whether it is possible to change conflict processing by implementing differential response-reward contingencies in congruent and incongruent trials (i.e., congruent, equal, and incongruent reinforced contingency). Experiments 1 and 2 revealed no evidence for significant adjustments in response to these differential contingencies–except for a trend-level effect of contingency in Experiment 1 indexing numerically faster responses in blocks in which congruent trials were rewarded more. However, we suspected that the absence of significant adjustment was maybe due to the adaptive threshold used in experiments 1 and 2. Results from an across experiment analysis conducted on the first sub-block of Congruent and Incongruent Reinforced contingency blocks revealed a change in conflict processing in the expected direction, and thus support our assumption. Consistent with this exploratory analysis, when implementing a fixed response-time threshold and only contrasting congruent and incongruent reinforced contingency blocks (experiment 3), we observed an RT adjustment in the form of a smaller congruency effect for blocks where incongruent trials were rewarded more. In addition, causality ratings suggested that participants correctly identified the manipulated contingencies, and the size of response adjustment between the two blocks was positively correlated with the differential ratings of the two contingencies.

The results of Experiment 3 suggest that it is possible to modulate the size of the congruency effect by selectively rewarding responses following congruent versus incongruent trials. This finding is consistent with a recent study by Chen et al., [45], who demonstrated increased Simon effect when performance-contingent reward always followed congruent trials, and inversely decreased Simon effect when reward followed incongruent trials. The results of Experiment 3 are also conceptually consistent with the evidence of selective reinforcement of task-switching found by Braem [26], and with recent propositions that the allocation and adjustment of cognitive control critically depends on a comparison between the reward prospect and the amount of (cognitive) effort needed to get that reward [14, 15]. In addition, the results extend a growing body of evidence showing an effect of instrumental-contingency on performance in tasks involving cognitive control [e.g., 29–31], and is consistent with an extensive literature from the associative learning domain [32–34]. This supports learning perspectives of cognitive control, according to which previous experiences learned through (reinforcement) learning processes guide the appropriate level of control that must be allocated to different contexts [16–18].

Our findings of Experiment 1 and 2 (in contrast to Experiment 3) highlight that there are limits with regard to the effect of differential reinforcement on congruent and incongruent trials. On the one hand, an adaptive reaction time threshold and/or introduction of an equally reinforced block might be sufficient to abolish the adaptive adjustment to congruency-reward contingencies. On the other hand, a cue signaling the identity of the next color-word stimulus (Experiment 2) was not sufficient to promote such adjustments. Finally, considering Experiment 3 alone, we did not find evidence for a generalization of condition-specific adjustments

to test items. Therefore, the effect of differential reward-contingencies observed in Experiment 3 could be more stimulus- than congruency-governed, and interrogate on the selective reinforcement of conflict processing representations and not (only) of task representations. However, these boundary conditions do not discredit the main observations in Experiment 3. We believe our experiments actually offer important pointers for the future investigation of instrumental contingency on cognitive control.

First, the evidence of differential reinforcement only for a fixed response-time threshold but not for an adaptive threshold is not completely surprising if we consider the well-known finding that only contingent but not non-contingent reward produces performance improvement, as well as the rationale of the EVC model. In experiments 1 and 2, the adaptive threshold allowed to maintain a constant reward rate across blocks and participants of .50. Even if it was possible for participants to locally expect higher reward prospect for performing fast and correct response, globally reward presentation was independent (or non-contingent) to the response (speed) performed by participants. Thus, there was no reason to allocate more cognitive control considering that the global reward expectancy was not influenced by the response performed. Consistent with this interpretation, the exploratory across experiment analysis revealed a response adjustment in the expected direction when the first sub-block of Congruent and Incongruent Reinforced contingency blocks is considered. More importantly, this contrasts with the differential reinforcement schedule of Experiment 3 with a fixed response-time threshold in which the gain of reward depends directly on response speed and accuracy, and where we found control adaptation. Future studies could systematically investigate not only the effect of instrumental contingency on cognitive control, but also how control processes are influenced by the probability of performing a "target" response, or response efficacy [46]. In addition, the use of a fixed time threshold in Experiment 3 was associated with a larger reward rate compared to Experiments 1 and 2. It is possible that this higher rate played a role in the strategic adaptation of conflict processing measured in Experiment 3 but not in the two first experiments. Thus, future studies could systematically investigate whether the effect of congruency-reward contingencies on conflict processing is modulated by reward rate.

Second, Experiment 2 suggests that the addition of a cue before a color-word stimulus was not sufficient to help pick up the reward-congruency contingencies and/or adjust their performance based on this information. Only congruency-reward contingencies seem to have modulated conflict processing in our study. This result is surprising considering the abundant evidence of performance modulations and modulated control processes using trial-by-trial cues [e.g., 10, 19]. However, our result echoes with a recent series of experiments by Jiménez et al. [47], in which the authors investigated the boundary conditions in which congruency cues can effectively influence (positively) control processes, and found a benefit of such cues only when cueing was deterministic, presented in between trials, with long intervals between trials, and a non-arbitrary stimulus-response mapping. Thus, a positive modulation of cognitive control processes by pre-cues seems to be largely dependent on the cue-contingency conditions manipulated. Similarly, there is a long tradition of research on the conditions for predictive stimulus-outcome learning, also referred to as Pavlovian learning [e.g., 42, 48], assuming that stimulus-outcome learning depends on reward prediction errors [49, 50, but see 51]. Thus, while recent studies (including ours) focused on the effect of instrumental contingency on tasks involving cognitive control, future studies might investigate the effect of advance reward or congruency information (cueing) considering the established boundary conditions from stimulus-outcome learning literature, to get a global understanding of the conditions and the processes that support the adaptation of simple responses and of cognitive control processes based on predictive cues.

Regarding the absence of a generalization to test items in Experiment 3, it is possible that participants adapted their responses to the different equally reinforced contingency implemented on test items, with equal reward rate on both congruent and incongruent test items, preventing the rule-governed generalization from learning to test items. In a future experiment, it might be interesting to use another method to test for generalization. One possibility would be to use a method with test stimuli presented in a separate block in the absence of reward. This procedure would prevent new learning caused by the equal-reward contingencies on congruent and incongruent items. Another possibility could be to work with designs that can contrast congruency conditions using a new stimulus on each trial [26, 52], thereby avoiding stimulus- or feature-specific reward effects. Finally, considering the subtle effect of congruency-reward contingency found on learning items and the similarity in the descriptive data between learning and test items, it is also possible that our experiment had a power problem that prevented us from detecting an effect on test items. Future investigations on the effect of congruency-reward contingencies will have to take this possibility into account.

In sum, by showing that it is possible to change conflict processing with condition-specific response-reward contingencies, the present study gives new important insights on the influence of reward on cognitive control. Our results are conceptually consistent with recent computational models of cognitive control (e.g., the EVC model; [14]) and support the learning perspective on cognitive control [16]. In addition, by highlighting boundary conditions of the effect of response-reward contingencies, we strongly believe that the present experiments offer new directions for the future investigation of instrumental contingency on cognitive control.

## Supporting information

**S1 Appendix. Instructions presented to the participants during the experiments.** (DOCX)

## Author Contributions

**Conceptualization:** Arthur Prével, Ruth M. Krebs, Senne Braem.

**Data curation:** Arthur Prével.

**Formal analysis:** Arthur Prével, Ruth M. Krebs, Nanne Kukkonen, Senne Braem.

**Funding acquisition:** Ruth M. Krebs, Senne Braem.

**Investigation:** Arthur Prével.

**Methodology:** Arthur Prével, Ruth M. Krebs, Senne Braem.

**Project administration:** Ruth M. Krebs, Senne Braem.

**Resources:** Arthur Prével.

**Software:** Arthur Prével.

**Supervision:** Ruth M. Krebs, Senne Braem.

**Validation:** Arthur Prével, Ruth M. Krebs, Senne Braem.

**Visualization:** Arthur Prével, Ruth M. Krebs, Senne Braem.

**Writing – original draft:** Arthur Prével, Ruth M. Krebs, Senne Braem.

**Writing – review & editing:** Arthur Prével, Ruth M. Krebs, Nanne Kukkonen, Senne Braem.

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
