## [Decision Letter · Decision Letter 0]

22 May 2021

PONE-D-21-10868

Selective reinforcement of conflict processing in the Stroop task

PLOS ONE

Dear Dr. PREVEL,

Thank you for submitting your manuscript to PLOS ONE. I have received two expert reviews and am ready to act. As you will see, the reviews of your manuscript are excellent and very relevant (thank you to both reviewers). The good news is that both reviews are positive and ask for minor revisions. My own reading of your manuscript converges with their decision. I will not spend time summarizing the reviews in any detail, you can see the detailed reviews below. Given these positive reviews, I invite you to submit a revised version of the manuscript that addresses the points raised during the review process.

Please pay attention to the following comments and give them due consideration, as those changes are required for acceptance. 

We look forward to receiving your revised manuscript.

Kind regards,

Ludovic Ferrand, Ph.D.

Academic Editor

PLOS ONE

Journal Requirements:

3. Please change "female” or "male" to "woman” or "man" as appropriate, when used as a noun (see for instance https://apastyle.apa.org/style-grammar-guidelines/bias-free-language/gender).

4. Please remove your figures from within your manuscript file, leaving only the individual TIFF/EPS image files, uploaded separately.  These will be automatically included in the reviewers’ PDF.

Reviewers' comments:

Reviewer's Responses to Questions

**Comments to the Author**

1. Is the manuscript technically sound, and do the data support the conclusions?

Reviewer #1: Yes

Reviewer #2: Yes

2. Has the statistical analysis been performed appropriately and rigorously? 

Reviewer #1: Yes

Reviewer #2: Yes

3. Have the authors made all data underlying the findings in their manuscript fully available?

Reviewer #1: Yes

Reviewer #2: Yes

4. Is the manuscript presented in an intelligible fashion and written in standard English?

Reviewer #1: Yes

Reviewer #2: Yes

5. Review Comments to the Author

Reviewer #1: I found this study interesting. It investigated the learning effect of reward on conflict processing in 3 experiments using the prime Stroop task. A performance-contingent reward was selectively presented after incongruent or congruent trials, leading to 3 types of congruency-reward contingencies: rewarded incongruent (RI) condition, rewarded congruent (RC) condition, and neutral condition (half incongruent and half congruent trials were rewarded). In Exp 1, a cue presented in the beginning of a trial to indicate the start of the trial. In Exp 2 and 3(?), the shape of the cue additionally indicated the congruency of the upcoming trial. In Exp 1 and 2, responses that were correct and faster than an adaptive threshold were rewarded. In Exp 3, responses that were correct and faster than a fixed threshold were rewarded. The results showed that in Exp 1 and 2, the first half block of data showed an interaction of congruency and contingency, and Exp 3 showed a similar interaction, indicating a smaller Stroop effect in the RI than in the RC condition. The authors interpret the results as that reward reinforced conflict processing in the RI condition and therefore reduced the conflict effect. I think the experimental design and analyses were generally sound, but the interpretation of the results needs some consideration. Below my comments are only minor.

1) As the shape of the cue in Exp 2 and 3(?) also indicated the congruency of the upcoming trial, participants may alternatively learn a shape-reward (instead of congruency-reward) contingency. This left open the possibility that the shape of the cue may modulate motivation trial-by-trial, and the increased motivation may enhance cognitive control in incongruent trials in the RI condition and therefore reduce the conflict effect. This alternative explanation might deserve some discussion.

2) In Exp 1 and 2, only the first half of the data showed the interaction of congruency and contingency. The authors suggest that this is because of the adaptive threshold. This is somehow speculation. Is there any way to analyze the data (e.g., comparing the first and the second data) to support this idea? In addition, learning reward contingency often takes time. Are participants able to learn and already utilize the congruency-reward contingency in the first half data?

3) What are the actual reward frequencies for the 3 experiments? I expect a frequency of 50% for Exp 1 and 2, and a higher frequency (> 50%) for Exp 3. Therefore, the stronger reward effect in Exp 3 could also be due to more rewarded trials?

Reviewer #2: In the manuscript „Selective reinforcement of conflict processing in the Stroop task” the authors present a series of three experiments aimed at investigating whether participants would adapt conflict processing to selective reinforcement of conflict (incongruent) or no-conflict (congruent) trials. Results show some indication of a reduced Stroop effect with selective reinforcement of performance in incongruent trials. But this effect seems to rely on specific boundary conditions.

The authors investigate a well-motivated research question and present interesting results. I have a few comments – aimed at improving the well written manuscript further – that prevent me from recommending publication of this manuscript in its present form.

- My first comment concerns the issue of contingency which has also been addressed by the authors themselves to some extent. In Experiments 1 and 2, the authors used a reward rate of .5 in all blocks which could mean that participants perceived the reward more like a random gain than a performance-contingent reward. (Perceived) reward contingency is associated with very different effects on cognitive control functions (Fröber & Dreisbach, 2014, 2016; Müller, Dreisbach, Goschke, Hensch, Lesch & Brocke, 2007) in studies investigating the effect of pre-cued reward. But also studies that used mere reward feedback (like in the present study) demonstrated oppositional effects of contingent and random reward on the congruency sequence effects (Braem, Verguts, Roggeman & Notebaert, 2012; Stürmer, Nigbur, Schacht & Sommer, 2011; van Steenbergen, Band & Hommel, 2009, 2012): Conflict adaptation was increased following a contingent reward but reduced following a random reward. This made me wonder, whether an additional exploratory analysis of the congruency sequence effect with the same design as in the across experiment analysis on pp. 17/18 could give more insight here. If it is indeed the case that rewards were perceived as less contingent in the second half of the blocks, one might see differences in the congruency sequence effects between first and second half.

- The fact that no significant effects with the factor time-on-task were found in the separate analyses of Experiments 1 and 2 and that a significant effect was found when collapsing data across experiments and focusing on the first half of blocks only, made me wonder about a possible power problem in this study. Maybe the influence of reward contingency on the Stroop effect is a rather subtle effect that needs more power to be detected with a high probability. Likewise, in Experiment 3 the critical interaction between Contingency and Congruency just falls below the conventional threshold of significance (p = .047) for the learning items and misses significance for the test items, while the descriptive data (Figure 5) looks pretty similar. Maybe the less pronounced interaction effect in test items would also need a higher-powered study to reach significance.

- My last comment might just be personal preference, but I would recommend to use a different naming for the different levels of the Contingency factor. While reading, Negative and Positive were not intuitive names for me. I understand that these names are a consequence of the formula explained on pp. 5/6, but the formula seems like an arbitrary choice to me (∆p could also be P(r|c) – P(r|i)). Therefore, I suggest to use more meaningful terms instead. For example, incongruent reinforced vs. congruent reinforced.

6. PLOS authors have the option to publish the peer review history of their article (what does this mean?). If published, this will include your full peer review and any attached files.

Reviewer #1: No

Reviewer #2: No

---

## [Author Response · Author response to Decision Letter 0]

6 Jul 2021

In what follows, we would like to respond to the individual remarks from each reviewer. Our responses to reviewers' comments are preceded by a dash. Changes to our manuscript are cited throughout this response letter and are highlighted in the revised manuscript file by track change.

Editor

- We changed the format of our manuscript according to PLOS ONE’s style requirements.

- We reviewed our reference list to ensure that it is complete and correct. 

Please change "female” or "male" to "woman” or "man" as appropriate, when used as a noun (see for instance https://apastyle.apa.org/style-grammar-guidelines/bias-free-language/gender).

- We changed the terms “female” and “male” to “woman” and “man” (lines: 140, 311, and 475).

Please remove your figures from within your manuscript file, leaving only the individual TIFF/EPS image files, uploaded separately. These will be automatically included in the reviewers’ PDF.

- The figures have been removed from the manuscript file and were uploaded separately.

Please include captions for your Supporting Information files at the end of your manuscript, and update any in-text citations to match accordingly. Please see our Supporting Information guidelines for more information: http://journals.plos.org/plosone/s/supporting-information.

- Caption for the Supporting Information file is now included at the end of our manuscript.

Reviewer 1

I found this study interesting. […] I think the experimental design and analyses were generally sound, but the interpretation of the results needs some consideration. Below my comments are only minor.

- We are thankful for the Reviewer's positive evaluation and helpful suggestions.

1) As the shape of the cue in Exp 2 and 3(?) also indicated the congruency of the upcoming trial, participants may alternatively learn a shape-reward (instead of congruency-reward) contingency. This left open the possibility that the shape of the cue may modulate motivation trial-by-trial, and the increased motivation may enhance cognitive control in incongruent trials in the RI condition and therefore reduce the conflict effect. This alternative explanation might deserve some discussion.

- We would like to thank the reviewer for raising this point, as it made us realize that the description of the procedure in Experiment 3 was probably not clear regarding the use of pre-cues. Cues that announce the congruency identity of the upcoming word were used only in Experiment 2, but not in Experiment 3. Consequently, the modulation of conflict processing can only be attributed, we believe, to the effect of congruency-reward contingencies. We now changed the description of the procedure in Experiment 3 (line 487) to clarify that no congruency cues were used in this experiment:

- “Thus, a unique geometric shape (circle or triangle, randomized across participants) was presented to signal a new trial to the participants, but not the upcoming congruency identity of the next color-word stimulus.”

- Second, we now also elaborated a bit more the discussion about the absence of this beneficial effect of congruency cues in Experiment 2 (line 703):

- “Second, Experiment 2 suggests that the addition of a cue before a color-word stimulus was not sufficient to help pick up the reward-congruency contingencies and/or adjust their performance based on this information. Only congruency-reward contingencies seem to have modulated conflict processing in our study. This result is surprising considering the abundant evidence of performance modulations and modulated control processes using trial-by-trial cues (e.g., Padmala & Pessoa, 2011; Van den Berg et al., 2014).”

2) In Exp 1 and 2, only the first half of the data showed the interaction of congruency and contingency. The authors suggest that this is because of the adaptive threshold. This is somehow speculation. Is there any way to analyze the data (e.g., comparing the first and the second data) to support this idea? In addition, learning reward contingency often takes time. Are participants able to learn and already utilize the congruency-reward contingency in the first half data?

- We agree that our assumption that this data pattern is caused by the adaptive threshold is speculative. We now also further highlighted this on line 443 (new text underscored):

- “Although speculative, we do believe this observation is in line with our assumption that the adaptive threshold (combined probably with a low reward rate) was responsible for the absence of a global adaptation of conflict processing in the expected direction.”

- As suggested by the reviewer, we now also ran the across-experiment analysis with Time-on-task as an additional within-subjects factor. This 2*2*2*2 repeated-measure ANOVA revealed (on RTs) a near-significant three-way interaction between Contingency, Congruency, and Time-on-task (p = .064). Together with the significant interaction between Contingency and Congruency found when only the first half was analyzed, this result further suggests that the pattern was specific to the first half of the block. This is consistent with our suggestion that the adaptive threshold used in Experiments 1 and 2 might have had a negative effect on performance and prevented a global adaptation of conflict processing in the expected direction. This fits with the idea that participants detected and adjusted control processes to the congruency-reward contingencies, but this effect decreases with time, contrasting with the results found in Experiment 3. This analysis is now added to the manuscript on page 18:

- “The analysis of Time-on-task in a 2*2*2*2 repeated measures ANOVA revealed no interaction between Contingency and Time-on-task (F(1, 82) = .014, p = .907, η_p^2 < .001), a near-significant interaction between Contingency, Congruency, and Time-on-task (F(1, 82) = 3.516, p = .064, η_p^2 = .041), and no interaction between Contingency, Congruency, Time-on-task, and Experiment (F(1, 82) = .032, p = .859, η_p^2 < .001). This result is in line with our assumption that the effect of contingency on conflict processing was only effective in the first half of the blocks due to the adaptive threshold used in Experiments 1 and 2.”

3) What are the actual reward frequencies for the 3 experiments? I expect a frequency of 50% for Exp 1 and 2, and a higher frequency (> 50%) for Exp 3. Therefore, the stronger reward effect in Exp 3 could also be due to more rewarded trials?

- Thank you for noting this. We agree this is valuable information and now added the reward rates for the three experiments to the manuscript (lines: 137, 312, and 476). The rates for Experiments 1, 2, and 3, were .213, 219, and .354, respectively. The results of .213 and .219 are consistent with the designs of Experiments 1 and 2, as only the half of trials were rewarded per block (e.g., the congruent trials only in the Negative/Congruent Reinforced block), and from those trials the adaptive threshold ensured that only 50% of responses were below the threshold and reinforced (i.e., around 25% of responses in total). The result of .354 in Experiment 3 is consistent with our assumption that a fixed-threshold would increase the reward rate compared to Experiments 1 and 2. We agree with Reviewer 1 that the effect of congruency-reward contingency found in Experiment 3 could be also due to the increased reward rate observed in this experiment. We already hinted at this in the original version of our manuscript, but now further discuss this possibility on lines 695-700:

- “In addition, the use of a fixed time threshold in Experiment 3 was associated with a larger reward rate compared to Experiments 1 and 2. It is possible that this higher rate played a role in the strategic adaptation of conflict processing measured in Experiment 3 but not in the two first experiments. Thus, future studies could systematically investigate whether the effect of congruency-reward contingencies on conflict processing is modulated by reward rate.”

 

Reviewer 2

The authors investigate a well-motivated research question and present interesting results. I have a few comments – aimed at improving the well written manuscript further – that prevent me from recommending publication of this manuscript in its present form.

- We would like to thank the reviewer for their positive remarks and helpful comments.

My first comment concerns the issue of contingency which has also been addressed by the authors themselves to some extent. In Experiments 1 and 2, the authors used a reward rate of .5 in all blocks which could mean that participants perceived the reward more like a random gain than a performance-contingent reward. (Perceived) reward contingency is associated with very different effects on cognitive control functions (Fröber & Dreisbach, 2014, 2016; Müller, Dreisbach, Goschke, Hensch, Lesch & Brocke, 2007) in studies investigating the effect of pre-cued reward. But also studies that used mere reward feedback (like in the present study) demonstrated oppositional effects of contingent and random reward on the congruency sequence effects (Braem, Verguts, Roggeman & Notebaert, 2012; Stürmer, Nigbur, Schacht & Sommer, 2011; van Steenbergen, Band & Hommel, 2009, 2012): Conflict adaptation was increased following a contingent reward but reduced following a random reward. This made me wonder, whether an additional exploratory analysis of the congruency sequence effect with the same design as in the across experiment analysis on pp. 17/18 could give more insight here. If it is indeed the case that rewards were perceived as less contingent in the second half of the blocks, one might see differences in the congruency sequence effects between first and second half.

- Thank you for this suggestion, which, we agree, is particularly interesting in the context of our study. However, we are afraid that the designs used in our study are not appropriate to test this effect. Specifically, testing the congruency sequence effect and its interaction with reward would require us to compare previous rewarded (versus non-rewarded) trials on (previous) congruent versus incongruent items. However, in the Negative/Congruent Reinforced and Positive/Incongruent Reinforced contingencies used in our study, reward was presented selectively on congruent trials or incongruent trials only. Only on the test items in Experiment 3, rewards were presented equally for both congruent and incongruent items. Unfortunately, however, also there we do not have the sufficient number of trials required to perform an analysis with enough statistical power (after calculation, 11 participants are found with at least one empty cell). Therefore, despite our shared interest in this analysis, we are afraid we were not able to perform it.

The fact that no significant effects with the factor time-on-task were found in the separate analyses of Experiments 1 and 2 and that a significant effect was found when collapsing data across experiments and focusing on the first half of blocks only, made me wonder about a possible power problem in this study. Maybe the influence of reward contingency on the Stroop effect is a rather subtle effect that needs more power to be detected with a high probability. Likewise, in Experiment 3 the critical interaction between Contingency and Congruency just falls below the conventional threshold of significance (p = .047) for the learning items and misses significance for the test items, while the descriptive data (Figure 5) looks pretty similar. Maybe the less pronounced interaction effect in test items would also need a higher-powered study to reach significance.

- We definitely agree with this comment, and particularly concerning the absence of significant interaction between Contingency and Congruency found on test items in Experiment 3. Also in our original manuscript, we were cautious to conclude that we currently “did not find evidence for a similar modulation on test items”, implicit in this sentence being that we certainly also do not find evidence against a modulation on test items. We now included this possibility more explicitly in the discussion (line 730):

- “Finally, considering the subtle effect of congruency-reward contingency found on learning items and the similarity in the descriptive data between learning and test items, it is also possible that our experiment had a power problem that prevented us from detecting an effect on test items. Future investigations on the effect of congruency-reward contingencies will have to take this possibility into account.”

My last comment might just be personal preference, but I would recommend to use a different naming for the different levels of the Contingency factor. While reading, Negative and Positive were not intuitive names for me. I understand that these names are a consequence of the formula explained on pp. 5/6, but the formula seems like an arbitrary choice to me (∆p could also be P(r|c) – P(r|i)). Therefore, I suggest to use more meaningful terms instead. For example, incongruent reinforced vs. congruent reinforced.

- Thank you for this suggestion. We agree with Reviewer 2 that our previous terminology was suboptimal. In this revised version of the manuscript, we replaced the terms Negative, Neutral, and Positive blocks by Congruent Reinforced, Equally Reinforced, and Incongruent Reinforced, respectively.

---

## [Decision Letter · Decision Letter 1]

16 Jul 2021

Selective reinforcement of conflict processing in the Stroop task

PONE-D-21-10868R1

Dear Dr. PREVEL,

I am pleased to inform you that your manuscript has been judged scientifically suitable for publication and will be formally accepted for publication once it meets all outstanding technical requirements.

Kind regards,

Ludovic Ferrand, Ph.D.

Academic Editor

PLOS ONE

Additional Editor Comments (optional):

Reviewers' comments:

Reviewer's Responses to Questions

**Comments to the Author**

1. If the authors have adequately addressed your comments raised in a previous round of review and you feel that this manuscript is now acceptable for publication, you may indicate that here to bypass the “Comments to the Author” section, enter your conflict of interest statement in the “Confidential to Editor” section, and submit your "Accept" recommendation.

Reviewer #1: All comments have been addressed

Reviewer #2: All comments have been addressed

2. Is the manuscript technically sound, and do the data support the conclusions?

Reviewer #1: Yes

Reviewer #2: Yes

3. Has the statistical analysis been performed appropriately and rigorously? 

Reviewer #1: Yes

Reviewer #2: Yes

4. Have the authors made all data underlying the findings in their manuscript fully available?

Reviewer #1: Yes

Reviewer #2: Yes

5. Is the manuscript presented in an intelligible fashion and written in standard English?

Reviewer #1: Yes

Reviewer #2: Yes

6. Review Comments to the Author

Reviewer #1: (No Response)

Reviewer #2: (No Response)

7. PLOS authors have the option to publish the peer review history of their article (what does this mean?). If published, this will include your full peer review and any attached files.

Reviewer #1: No

Reviewer #2: No

---

## [Editor Report · Acceptance letter]

22 Jul 2021

PONE-D-21-10868R1 

Selective reinforcement of conflict processing in the Stroop task 

Dear Dr. Prével:

I'm pleased to inform you that your manuscript has been deemed suitable for publication in PLOS ONE. Congratulations! Your manuscript is now with our production department. 

Kind regards, 

on behalf of

Dr. Ludovic Ferrand 

Academic Editor

PLOS ONE